# socialh: An R package for determining the social hierarchy of animals using data from individual electronic bins

Júlia de Paula Soares Valente[1,2]*, Matheus Deniz[2], Karolini Tenffen de Sousa[2], Maria Eugênia Zerlotti Mercadante[3], Laila Talarico Dias[1,2]

1 Laboratório de Genética Aplicada ao Melhoramento Animal (GAMA), Departamento de Zootecnia, Universidade Federal do Paraná (UFPR), Curitiba, Paraná, Brazil, 2 Programa de Pós-Graduação em Zootecnia–UFPR, Curitiba, Paraná, Brazil, 3 Instituto de Zootecnia, Centro Avançado de Pesquisa de Bovinos de Corte, Sertãozinho, São Paulo, Brazil

* juliadepaulasoaresvalente@gmail.com

**Data Availability Statement:** The database used in the examples is available in socialh package (https://cran.r-project.org/web/packages/socialh/index.html) and Kaggle repository, (https://www.

## Abstract

Cattle have a complex social organization, with negative (agonistic) and positive (affiliative) interactions that affect access to environmental resources. Thus, the social behaviour has a major impact on animal production, and it is an important factor to improve the farm animal welfare. The use of data from electronic bins to determine social competition has already been validated; however, the studies used non-free software or did not make the code available. With data from electronic bins is possible to identify when one animal takes the place of another animal, i.e. a replacement occurs, at the feeders or drinkers. However, there is no package for the R environment to detect competitive replacements from electronic bins data. Our general approach consisted in creating a user-friendly R package for social behaviour analysis. The workflow of the socialh package comprises several steps that can be used sequentially or separately, allowing data input from electronic systems, or obtained from the animals' observation. We provide an overview of all functions of the socialh package and demonstrate how this package can be applied using data from electronic feed bins of beef cattle. The socialh package provides support for researchers to determine the social hierarchy of gregarious animals through the synthesis of agonistic interactions (or replacement) in a friendly, versatile, and open-access system, thus contributing to scientific research.

## Introduction

The automation of production systems has allowed the simultaneous monitoring of various animals' parameters. For example, sensors have been shown to be useful for monitoring the cows' location [1], activities (e.g., walking [2, 3]; lying down [4, 5]), and feeding and drinking behaviour (time and duration [6, 7]). Furthermore, electronic feeders and drinkers are also useful for detecting social competition [8–10] since most disputes occur during feed time [11, 12] and at drinkers on hot days [13]. With the data from electronic bins is possible to define a

kaggle.com/datasets/juliavalente/data-from-visits-to-the-trough-of-nellore-cattle). The package can be downloaded from the Comprehensive R Archive Network (CRAN) version 0.1.0 a https://cran.r-project.org/web/packages/socialh/index.html. A development version, issue logging, and support can be found at (https://github.com/juliapsvalente/socialh).

**Funding:** We thank the São Paulo Research Foundation (FAPESP - https://fapesp.br/en; grant #2017/10630-2 and grant #2017/50339-5) for the financial support to perform data collection and to publish this article. The funders had no role in study design, data collection and analysis, decision to publish, or preparation of the manuscript.

**Competing interests:** The authors have declared that no competing interests exist.

competition; once an animal (the actor) takes the place of the previous animal (reactor) at the bin [8, 10].

Previous studies have applied algorithms for the autonomous detection of replacements using data from individual electronic bins [8–10, 13]. A replacement occurs when an animal that occupied the bin (reactor) is completely withdrawn by another animal that occupied the same bin (actor) in a short interval. However, there are no studies that assessed the general applicability of a replacement detection algorithm developed in open-access systems (e.g., R software) to assess social relationships, such as social hierarchy of farm animals. Thus, our general approach was to develop and to make available a user-friendly R package, named socialh, that can be used to detect replacements using data from individual electronic bins to determine the social hierarchy based on the competition between each pair of animals in the herd [14]. The aim of this study is to describe the socialh package version 0.1.0. Our package is intended for users who wish to implement more flexible (i.e. it is possible include data from electronics bins and data observational) social behaviour analysis since the R environment allows the integration of several functions from different packages. In addition, we provide an overview of socialh and demonstrate its features and applicability using data from electronic feed bins of beef cattle.

## Overview of socialh

Our general approach consisted of created a user-friendly R package for competition behaviour analysis. We choose to use the R software [15] because it is an open-access software and that it offers several resources for data analyses. The socialh package is available from the Comprehensive R Archive Network (CRAN) at https://cran.r-project.org/web/packages/socialh/index.html. The work-flow of the socialh package comprises several steps (Fig 1) that can be used sequentially or separately. First, we developed a function to identify replacements using data from electronic bins (feeder or drinker); a replacement occurs when an animal that occupied the bin (reactor) is completely withdrawn by another animal that occupied the same bin (actor) in a short interval. Second, we integrated other functions to determine the social rank and social hierarchy of the herd. The functions of the socialh package are listed in Table 1.

## Database

For the present article, we used a database obtained from feed efficiency test of beef cattle to illustrate the functions of the socialh package. The data were provided by the Beef Cattle Research Center, Institute of Animal Science, Sertãozinho, São Paulo State, Brazil. All management procedures followed animal welfare guidelines and were conducted in accordance with State Law No. 11 977 of the State of São Paulo, Brazil.

The database used in the examples is available on socialh package and Kaggle repository (https://www.kaggle.com/datasets/juliavalente/data-from-visits-to-the-trough-of-nellore-cattle).

The feed efficiency test was conducted in 2021, including 37 Nellore males with a mean age ± SD of 292 ± 26 days and a mean weight of 255.9 ± 44.5 kg. The group was housed for 21 days in a paddock (3138 m$^2$) containing 5 electronic feed bins (GrowSafe System$^®$, Vytelle–Kansas City, Missouri, USA). The Total Mixed Ration (TMR) was offered twice a day (8:00 and 15:00 hours) and the animals had *ad libitum* access to TMR and water trough. All animals were fitted with an ear tag transponder that allowed the electronic bin system to record the date and time when each animal entered and left the feeder. The database obtained from the electronic feeder consisted of 90,211 lines of feeding events. A feeding event starts when the

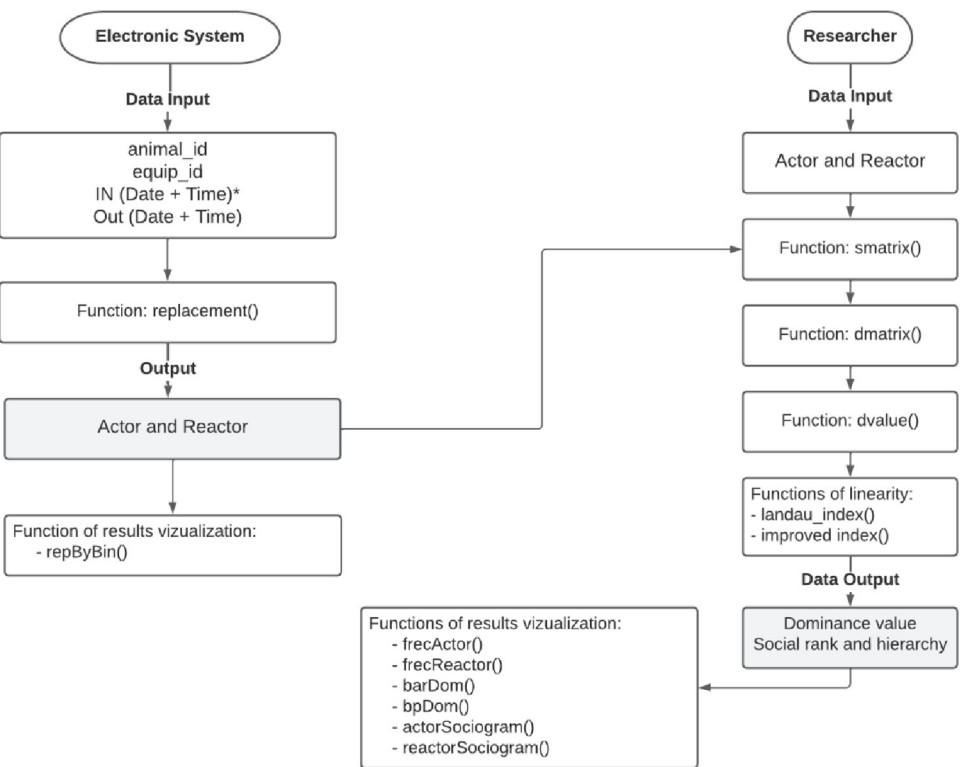

**Fig 1. R package flow chart for competition behaviour analysis.** *Format of datasheet: date—dd/mm/yyyy; time—hour:minutes:seconds.

ear tag of an animal α is recognized by a specific feeder β and ends when the animal α completely withdraws its head from the feeder β or when a different ear tag transponder of another animal is recognized by the feeder.

**Table 1. Descriptions of functions of the socialh R package.**

| Function | Description |
| --- | --- |
| replacement | Identifies replacements between the actor and reactor from electronic bin data. |
| repByBin | Identifies the frequency of replacements by bin. |
| freqActor | Identifies the frequency that an animal was actor. |
| freqReactor | Identifies the frequency that an animal was reactor. |
| smatrix | Builds a square matrix containing the frequency of competition between each dyad (each pair of animals). |
| dmatrix | Determines the $S_{ij}$ dyadic dominance relationship from a sociomatrix. |
| dvalue | Determines the dominance value, social rank and hierarchy from the $S_{ij}$ dyadic relationship matrix. |
| landau_index | Calculates the linearity index developed by Landau (1951). |
| improved_index | Calculates the linearity index improved by de Vries (1995). |
| barDom | Generates a barplot from the variables obtained in the dvalue function (dominance value, social hierarchy and social rank). |
| bpDom | Generates a boxplot from the variables obtained in the dvalue function (dominance value, social hierarchy and social rank), and variable obtained in the frequency functions (freqActor, and freqReactor). |
| actorSociogram | Generates a sociogram with actor information. |
| reactorSociogram | Generates a sociogram with reactor information. |

## Specifications and implementation of the functions

### Data preparation and import

We would like to highlight that the data preparation depends on the database and the objective of each research; we therefore did not include these functions in the package. The user has two ways to import the database: (1) data from electronic bins; (2) data from animals' observation. For the analysis of competition behaviour data from individual electronic bins must contain the equipment identification (column name: equip_id), animal identification (column name: animal_id), date and time (dd/mm/yyyy and hour:minutes:seconds) when the animal entered at the electronic bin (column name: IN), and date and time (dd/mm/yyyy and hour:minutes: seconds) when the animal left the electronic bin (column name: OUT). On the other hand, when competition behaviour is analysed using data collected by video or in person, the database must contain two parameters (columns): actor and reactor.

In our case, the data obtained by the electronic bins had the following columns: animal_id, equip_id, date and time of the animal entry into the bin, duration (s), and consumption (kg). To use the socialh package, we had to prepare the database. We determined the date and time when the animal left the bin by summing the date and time when the animal entered at the electronic bin and the duration of the feeding event (column "Duration(s)"). Thus, after prepared the database, we kept the columns with animal identification (animal_id), bin identification (equip_id), date and time of the entry into the bin (IN), and date and time of the exit from the bin (OUT).

### Replacement identification

The first function of the socialh R package, `replacement()`, is responsible to identify replacements at the electronic bins. In our package, we used the definition of replacement proposed by Huzzey et al. [8]. A replacement was identified when the algorithm detected a specific time-interval between two different animals that sequentially visited the same electronic bin (feed or water), i.e., the animal that occupied the electronic bin (reactor) was completely withdrawn by the following animal that occupied the same electronic bin (actor). Thus, before the user apply the `replacement()` function, it is necessary to determine the optimum time-interval to the algorithm identify a replacement.

The definition of the optimal time-interval depends on the species and animal category. For example, for lactating Holstein cows, previous studies used data from feeders and water bins to determine different intervals for the identification of a replacement. Huzzey et al. [8] highlighted that, for feed bins, a shorter interval ($\leq$26 s) between successive feeding events of two cows at one feed bin was associated with competitive replacement. On the other hand, for water bins, McDonald et al. [10] found the optimal time-interval for the identification of replacements to be $\leq$29 s. Combining electronic feed and water bin data, Foris et al. [9] found that a 20 to 30s interval was optimal to identify competitive replacements. For the database used as example in this study, we used the time-interval of 0-10s to identify the competitive replacement. As the objective of this study was to present the socialh package, we did not focus in determine the optimal time-interval for identify competitive replacement of Nellore cattle. When this study was performed, we were only aware of the available dairy cattle literature, so the choice of the range (0-10s) was based on the authors' previous experience with Nellore cattle. However, we strongly recommend that future studies determine the optimal time-interval to identify replacements in electronic bins of different breeds and categories of cattle.

After pre-processed the database from electronic bins and determinate the optimal time-interval to identify a competitive replacement, the user can apply the `replacement()`

| animal_id | equip_id | IN | OUT |
|---|---|---|---|
| 459 | 5 | 29/06/2021 09:24:32 | 29/06/2021 09:25:23 |
| 459 | 5 | 25/06/2021 15:44:20 | 25/06/2021 15:44:24 |
| 459 | 2 | 15/07/2021 09:50:04 | 15/07/2021 09:50:43 |
| 459 | 2 | 03/07/2021 15:33:35 | 03/07/2021 15:34:06 |
| 459 | 3 | 08/07/2021 15:21:55 | 08/07/2021 15:22:01 |
| 459 | 2 | 01/07/2021 18:00:58 | 01/07/2021 18:01:07 |
| 459 | 1 | 09/07/2021 09:10:59 | 09/07/2021 09:12:10 |
| 459 | 1 | 27/06/2021 16:35:40 | 27/06/2021 16:36:17 |

**Fig 2. A sample of the input database with data from the electronic feeding system of Nellore cattle used in the replacement() function.**

function specifying the following parameters: database—file containing four columns (equip_id, animal_id, IN and OUT); and sec—optimal time-interval in seconds, i.e., [replacement(database, sec.)]. In order to identify a replacement, the function replacement() will order the database according to the columns equip_id and IN (Fig 2). If the time interval between two animals is lower than the time specify at the function, the event will be recognized as a competitive replacement. The output data frame of the replacement() function is printed in two columns: actor and reactor. In our database example, the replacement function using the interval of 0-10s found 54,346 competitive replacements.

```
#First, install and load the socialh R package from CRAN
  repository

> install.packages("socialh")

> library(socialh)

#Load the database

> example.data <- read.csv("behaviour_data.csv")

# Apply the replacement(x, sec) function to create a data table
  with actor and reactor and save as an object to use later.

> replace <- replacement (example.data, 10)

> head(replace)

actor reactor

336 704

128 336

336 704
```

```
704 336
```

```
465 836
```

```
465 798
```

**Frequency determination.** The socialh package provides three functions that return frequency information: `repByBin()`, `freqActor()`, `freqReactor()`. The `repByBin()` function, like the `replacement()` function, uses the data from the electronic bin system (equip_id, animal_id, IN, and OUT) and the output is the frequency of replacements that occurred in each bin. The `freqActor()` and `freqReactor()` functions return the frequency that each animal was actor and reactor.

```
#Apply the repByBin(x, sec) function to create an output with
  frequency information of replacements that occurred in each
  bin.
```

```
>replacementByBin<- repByBin(example.data, 10)
```

```
>print(replacementByBin)
```

```
equip_id replacements %
```

1. 1 9216 16.94616

2. 2 12217 22.46433

3. 3 11788 21.67549

4. 4 10698 19.67123

5. 5 10465 19.24279

```
#Apply the freqActor(x) function to create an output with fre-
  quency of an animal was actor.
```

```
>fActor<- freqActor(replace)
```

```
>head(fActor)
```

```
animal_id freq_actor %
```

```
109 801 1.4738895
```

```
117 1302 2.3957605
```

```
128 1827 3.3617930
```

```
146 2760 5.0785706
```

```
181 1285 2.3644794
```

```
227 937 1.7241379
```

```
#The freqReactor(x) function create an output with frequency of
  an animal was reactor.
```

```
>fReactor<- freqReactor(replace)
```

```
>head(fReactor)
```

```
animal_id freq_actor %
```

```
109 802 1.475729585

117 1269 2.335038457

128 1899 3.494277408

146 2354 4.331505539

181 1558 2.866816325

227 1272 2.340558643
```

**Matrix determination: Sociometric and dyadic.**   The output data frame of replacement function (actor and reactor) permits to determine the dominance value for each animal. For the socialh package, we chose to use the method proposed by Kondo and Hurnik [14] to determine the dominance value. There are available several R packages and software that can be used to infer the animals' dominance value, but these software employ different methodologies. For example, the R package named "aniDom" uses the original and the randomized Elo-rating method [16]; the R package "Elorating" calculates David's scores [17]; the R package "compete" runs the I&SI method [18]; and the R package "steepness" also calculates David's scores and normalized David's scores [19]. The Elo-rating method considers the proportions of wins and losses of a dyad, where the rating of the winner of the dispute is increased by an amount that depends on the chance of winning: the amount is small if the chance of winning is high and vice versa [20]. The Elo-rating and David's score consider that the overall success of an individual is determined by weighting each dyadic success measure by the unweighted estimate of the inter-actant's overall success so that relative strengths of the other individuals are considered [21]. While the Kondo and Hurnik [14] method considers all competition than an animal was involved (i.e., as actor or reactor) in relation to other herd members, without ponderations. Other authors also preferred this index for similar reason [11, 12, 22–24]. To our knowledge, socialh is the first R package that adopted the method proposed by Kondo and Hurnik [14].

To obtain the dominance value based on Kondo and Hurnik [14], we developed three functions: `smatrix()`, `dmatrix()` and `dvalue()`. The functions, which are described below, can be used sequentially, or combined with functions from other packages that also determine the dominance value of an animal. We also highlight that the user can apply these three functions to analyse data obtained through the `replacement()` function or input data from direct (in-person) or indirect (video) observations. The `smatrix()` function builds a square matrix that contains the frequency of competition (replacements) between each dyad (each pair of animals). The `dmatrix()` function transforms the smatrix into a dyadic dominance relationship as proposed by Kondo and Hurnik [14]. Therefore, the dyadic dominance relationship of the $i$th animal relative to the $j$th animal ($S_{ij}$) is assessed qualitatively by the sign of the difference between $X_{ij}$ and $X_{ji}$, which always results in a value of -1, 0 or +1 (Eq 1). The values distinguish four relationships: domination (value +1), subordinations (value -1), tied (i.e., equal numbers of wins for both members of a dyad), and unknown (no data) relationship (both values 0).

$$S_{ij} = \frac{X_{ij} - X_{ji}}{|X_{ij} - X_{ji}|} \tag{1}$$

The output data frame of `smatrix()` is a square matrix with the actors in the column and reactors in the row (Fig 3A) and the output of `dmatrix()` is a square matrix containing the dyadic dominance relationship (Fig 3B).

```
#Use the smatrix() function to create sociomatrix by a replace-
  ment data table and save as an object to use later.
```

a)

|  | | Actor | | | | |
|---|---|---|---|---|---|---|
|  |  | A | B | C | D | E |
| Reactor | A | 0 | 0 | 1 | 0 | 2 |
|  | B | 2 | 0 | 0 | 2 | 0 |
|  | C | 0 | 3 | 0 | 2 | 0 |
|  | D | 2 | 2 | 1 | 0 | 3 |
|  | E | 5 | 0 | 1 | 4 | 0 |

b)

|  | | Actor | | | | |
|---|---|---|---|---|---|---|
|  |  | A | B | C | D | E |
| Reactor | A | 0 | -1 | +1 | 0 | -1 |
|  | B | +1 | 0 | -1 | 0 | 0 |
|  | C | -1 | +1 | 0 | +1 | -1 |
|  | D | +1 | 0 | -1 | 0 | -1 |
|  | E | +1 | 0 | +1 | +1 | 0 |

**Fig 3.** Example of the output data frame of the (a) `smatrix()` function and (b) `dmatrix()` function.

```
> social <- smatrix (replace)

> head(social)

Actor

reactor 109 117 128 146 181 227

109 0 1 1 1-1-1

117-1 0 1 1-1-1

128-1-1 0 1-1-1

146-1-1-1 0-1-1

181 1 1 1 1 0 1

227 1 1 1 1-1 0

#Apply the dmatrix() function to transform the sociomatrix in a
   dyadic dominance relationship matrix and save as an object to
   use later.

> dyadic <- dmatrix (social)

> head(dyadic)

equip_id replacements %

   1. 1 9216 16.94616

   2. 2 12217 22.46433

   3. 3 11788 21.67549

   4. 4 10698 19.67123

   5. 5 10465 19.24279
```

The `dvalue()` function sums the dyadic dominance relationship by column (actor) as proposed by Kondo and Hurnik [14]. Therefore, the dyadic relationship of the $i^{th}$ animal relative to the $j^{th}$ animal ($S_{ij}$) is assessed qualitatively according to Eq (2).

$$S_i = \sum_{j-i}^{n} S_{ij} \qquad (2)$$

where $S_i$ is the sum of all relationships involving animal *i*, and *n* is the number of possible interactions of one animal in the group with the other animals.

The social rank (high and low) and social hierarchy (dominant, intermediate, and subordinate) are determined according to dominance value. The choice of dividing the group according to social rank (2 categories) or social hierarchy (3 categories) depends on the study objectives. Both social rank (SR) and social hierarchy (SH) are estimated by the distance between the highest (+ X) and the lowest (- Y) dominance value, plus 1 (corresponds to the dominance value zero), which determines the number of points in the range (Eq 3; see [22]).

$$\text{SR or SH} = \frac{|\text{Distance between highest } (+X) \text{ and lowest } (-Y) \text{ dominance value}| + 1}{2 \text{ or } 3} \quad (3)$$

For social rank, animals with dominance values in the first half, i.e., those with the lowest values including negative ones, are classified as low rank, and animals with dominance values in the second half are classified as high rank. For social hierarchy, animals with dominance values in the first tertile, i.e., those with the lowest values including negative ones, are classified as subordinate. Animals with dominance values in the second tertile are classified as intermediate, and animals with dominance values in the third tertile with higher positive values are classified as dominant. The output data frame of the dvalue() function is printed in four columns: animal_id, dominance_value, social_rank, and social_hierarchy.

```
#Employ the dvalue() function to determine dominance value,
  social rank and social hierarchy by a dyadic matrix.

> dominance <- dvalue (dyadic, hs = TRUE, rs = TRUE)

> head(dominance)

animal_id dominance_value social_hierarchy social_rank

1: 227 -26 subordinate low

2: 426 -20 subordinate low

3: 757 -18 subordinate low

4: 181 -16 subordinate low

5: 764 -16 subordinate low

6: 975 -16 subordinate low

> tail(dominance)

animal_id dominance_value social_hierarchy social_rank

1: 288 15 dominate high

2: 737 16 dominate high

3: 980 17 dominate high

4: 146 18 dominate high

5: 787 19 dominate high

6: 834 18 dominate high
```

**Visualization of the results.** To visualize the results, the package provides functions based on the "ggplot2" [25] and "circlize" [26] packages. The barDom() function returns to

```
>barDom(dominance, dominance$social_hierarchy)
```

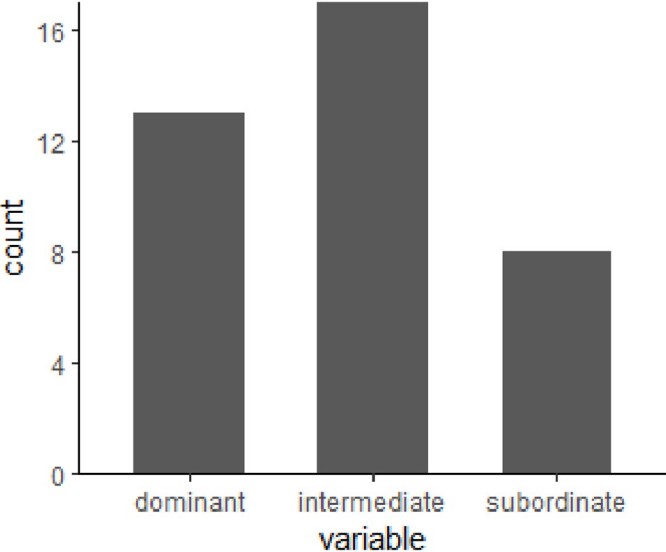

```
>bpDom(x, y)
```

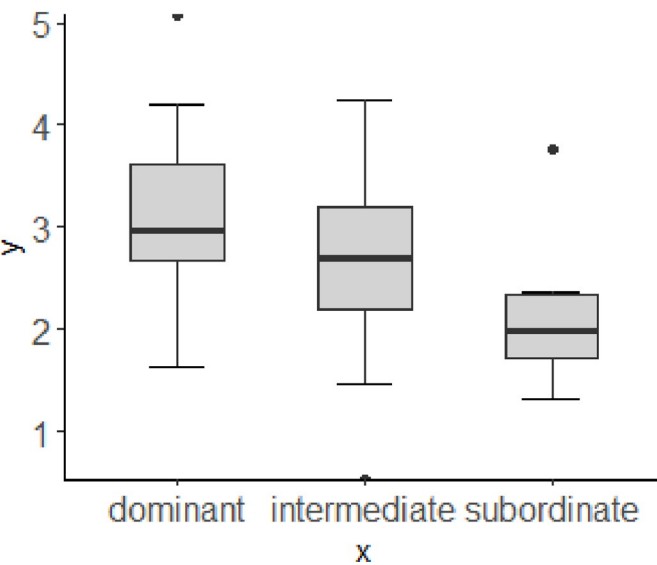

**Fig 4.** Example of barplot (above) and boxplot (below) using `barDom` and `bpDom` functions, respectively.

the user a barplot with information related to the values obtained by the `dvalue()` function. The purpose of this function is graphically demonstrating the number of animals in each social category social hierarchy (dominant, intermediate and subordinate) or social rank (high and low rank)) (Fig 4). To inspect the distribution, outliers, mean and standard deviation of the dominance values, the user can apply the `bpDom()` function to obtain a boxplot (Fig 4).

The sociometric matrix can be visualized through sociograms generated by the `actorSociogram()` and `reactorSociogram()` functions, which visually display the actor and reactor relationships between the animals within the evaluated group (Fig 5). The animals are represented around a circular plot and are connected by arrows in which the thickness of the

```
> actorSociogram(smatrix)
```

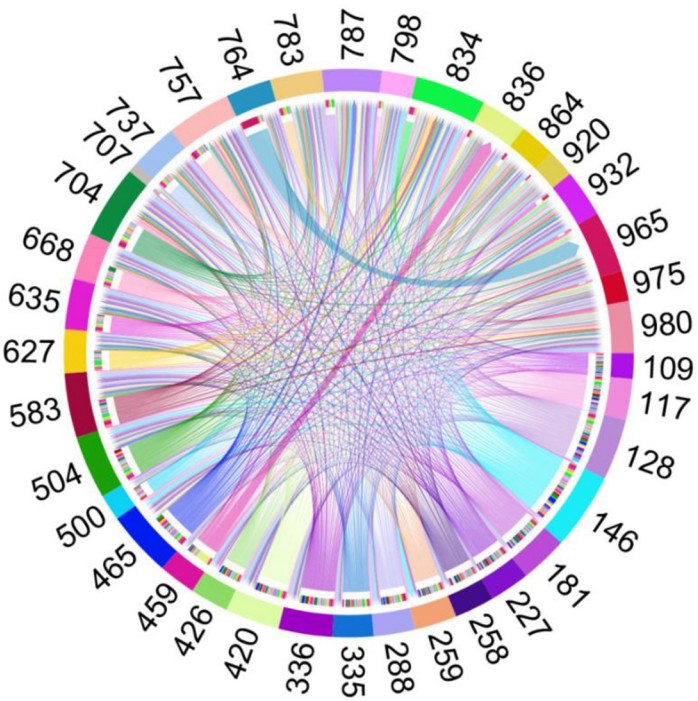

```
> reactorSociogram(smatrix)
```

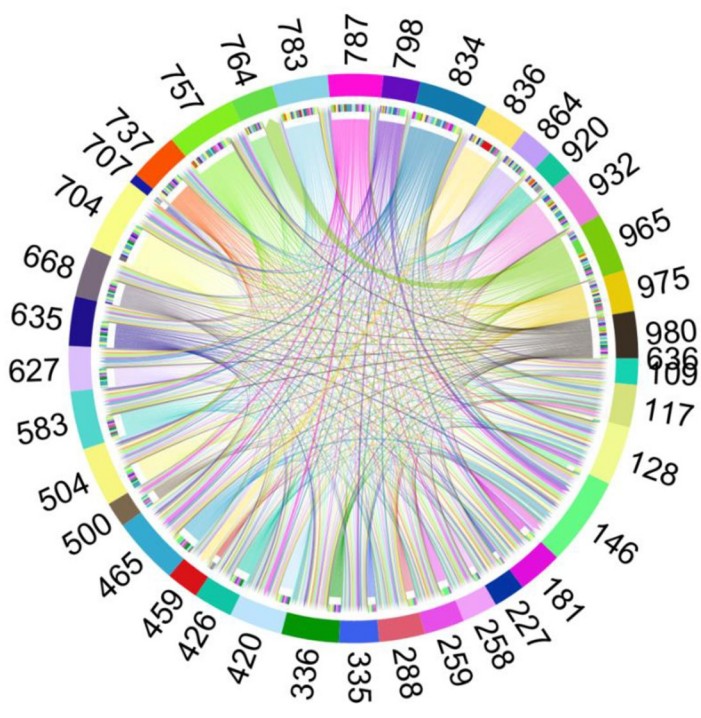

**Fig 5.** Example of sociograms using `actorSociogram` (above) and `reactorSociogram` (below) functions, respectively.

lines is proportional to the frequency of interactions between the animals and the arrowheads indicate the direction of the interactions.

**Linearity index.**   In social organization of animals, a hierarchy is considered linear when one animal A dominates all others in the group; while in a circular relationship, the animal A dominates B and C, B dominates C, that dominates D, and D dominates B. Also, unknown relationships can occur in a herd, this happens when no competitive behaviour is observed between a dyad; or when the number of winning and losses of a dyad are equal [27]. The linearity index indicates the number and effects of unknown relationships in a herd; as increase the number of unknown relationships, decreases the calculated linearity index [28].

First, we developed the `landau_index()` function to determine the linearity index (h; Eq 4) of the group as proposed by Landau [29]. Landau [29] has devised an index that measure the degree of linearity in a set of dominance relationships between the animals in the same group. The Landau index (h) use the concept of 'number of dominated animals', where a value of h > 0.9 generally indicates a strongly linear hierarchy [30]. To apply the `landau_index ()` function, the user must specify a dyadic matrix obtained with the `dmatrix()` function, i.e., [`landau_index (dmatrix)`].

$$h = \left(\frac{12}{n^3 - n}\right) \sum_{a=1}^{n} \left[ V_a - \left(\frac{n-1}{2}\right) \right]^2 \qquad (4)$$

where *h* is the linearity index; *n* is the number of animals in the group, and *V* is the number of animals that a given animal dominated.

```
#Apply the landau_index() function to determine the linearity
  index by a dyadic matrix.

> landau <- landau_index (dyadic)

> print(landau)

[1] 0.318397
```

Second, we developed the `improved_index()` function. This function applies the improved linearity test (h'; Eq 5) as described by de Vries [27]. de Vries [27], developed a modified linearity index (h′) that aims to correct for unknown and tied relationships. To apply the `improved_index()` function, the user must specify a dyadic matrix obtained with the dmatrix function and the sociomatrix obtained with the smatrix function, i.e., [`improved_index (dmatrix, smatrix)`]. Both indexes range from 0 to 1, with 0 being a non-linear (every animal in the group tends to dominate the same number of other animals) and 1 a perfectly linear hierarchy (an animal dominates all animals ranked below).

$$h' = h + \frac{6}{n^3 - n} * u \qquad (5)$$

where *h* is the Landau linearity index; *n* is the number of individuals, and *u* is the number of unknown relationships.

```
#Apply the improved_index() function to determine the improved
  linearity index by a dyadic matrix and a sociomatrix.
```

```
> improved_index <- landau_index (dyadic, social)

> print(improved_index)

[1] 0.331911
```

## Relationship between different methods for determining the dominance index available in R packages

There are R packages (e.g., "aniDom", "Eloranting", "steepness") that use different methods (Elo Rating, I&SI, and David Score) for determining the dominance index of gregarious animals, but none of these packages present the index proposed by Kondo and Hurnik [14]. Thus, we determined the dominance index from the different methods available in R packages for the same database of Nellore cattle used in the previous examples. Then, we perform Pearson's correlation analysis with p-value among the results from all methods including the Kondo and Hurnik index values from the socialh packages (Table 2).

The dominance index from the different methods were significantly correlated, except between Elo Rating and ISI. The correlations between the Kondo and Hurnik index and David Score and Elo Rating were positive, i.e., animals with highest values in one of the methods, also present high values in the other method. On the other hand, the I&SI showed negative correlations because the way in which the animals are classified by this method is opposite to that obtained by the other methods. In this case, the main difference between the methods is the interpretation of the results. In some cases, two animals could receive an identical index value, as occurred in the Kondo and Hurnik Index, and David Score. When applying the I&SI method, we noted that each animal obtained a single value which ranges from 1 to the total number of animals in the group (e.g., in our example ranged from 1 to 38). Another characteristic of the I&SI methods is present only positive values, while the other methods (David Score, Elo Rating, and Kondo and Hurnik Index) classify animals with values that vary from negative to positive. These findings indicate that the I&SI is the most discriminative and Kondo and Hurnik Index, and David Score are the least discriminative. Despite the correlations between values, each method has specificities that must be observed at the moment of application. Thus, the researcher should decide which index best suits to the group of animals that will be evaluated.

## Final considerations

The socialh R package provides support for researchers to determine the social hierarchy of gregarious animals through the synthesis of competition behaviour in a friendly, versatile, and open access software, thus contributing to scientific research. Limitations of socialh include

**Table 2. Correlations among dominance index for a Nellore cattle group obtained from different methods (Elo Rating, I&SI, and David Score) available on R packages.**

|  | Kondo and Hurnik Index | David score | Elo rating | I&SI |
|---|---|---|---|---|
| Kondo and Hurnik Index | 1 | 0.871*** | 0.291* | -0.641* |
| David score |  | 1 | 0.418*** | -0.553*** |
| Elo rating |  |  | 1 | -0.132 |
| I&SI |  |  |  | 1 |

*P < 0,10

***P < 0,01

the fact that the package uses only the method proposed by Kondo and Hurnik [14], which does not mean that researchers cannot adopt other methods or integrate the results obtained with socialh in other packages. Second, the `replacement()` function uses the interval-time between animals as a parameter to identify replacements at the bin; however, it is up to the user to inform the value that best suits to the specie studied. We encourage further studies to estimate the most adequate replacement times for different animal species and category. Finally, the socialh can help researchers on determining the social hierarchy of gregarious animals using programming languages.

## Acknowledgments

We thank the staffs of Beef Cattle Research Center, Institute of Animal Science, Sertãozinho, São Paulo State, Brazil for helping in the data acquisition. Also, we thank the professor Luiz Carlos Pinheiro Machado Filho, PhD, for all the shared knowledge that motivated us to work with social behaviour.

## Author Contributions

**Conceptualization:** Júlia de Paula Soares Valente, Matheus Deniz, Karolini Tenffen de Sousa.

**Data curation:** Maria Eugênia Zerlotti Mercadante.

**Funding acquisition:** Maria Eugênia Zerlotti Mercadante.

**Investigation:** Júlia de Paula Soares Valente.

**Methodology:** Matheus Deniz, Karolini Tenffen de Sousa.

**Resources:** Maria Eugênia Zerlotti Mercadante.

**Software:** Júlia de Paula Soares Valente, Laila Talarico Dias.

**Supervision:** Maria Eugênia Zerlotti Mercadante, Laila Talarico Dias.

**Writing – original draft:** Matheus Deniz, Karolini Tenffen de Sousa.

**Writing – review & editing:** Júlia de Paula Soares Valente, Maria Eugênia Zerlotti Mercadante, Laila Talarico Dias.

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
