## [Decision Letter · Decision Letter 0]

13 Dec 2021

PONE-D-21-29427

socialh : An R package for determining the social hierarchy of cattle using data from individual electronic bins

PLOS ONE

Dear Dr. Valente,

Thank you for submitting your manuscript to PLOS ONE. After careful consideration, we feel that it has merit but does not fully meet PLOS ONE’s publication criteria as it currently stands. Therefore, we invite you to submit a revised version of the manuscript that addresses the points raised during the review process.

The paper is interesting, however considered that is missing information and there are details that need to be clarified.

We look forward to receiving your revised manuscript.

Kind regards,

Arda Yildirim, Ph.D.

Academic Editor

PLOS ONE

“We thank the São Paulo Research Foundation (FAPESP -  https://fapesp.br/en; grant #2017/10630-2 and grant #2017/50339-5).

JPSV, MD, and KTS acknowledge the Coordination for the Improvement of High Education Personnel (CAPES - https://www.iie.org/programs/capes ) for the scholarship.”

3.Thank you for stating the following in the Acknowledgments Section of your manuscript:

“We thank the São Paulo Research Foundation (FAPESP, grant #2017/10630-2 and grant #2017/50339-5). We thank the staffs of Beef Cattle Research Center, Institute of Animal Science, Sertãozinho, São Paulo State, Brazil for helping in the data acquisition. JPSV, MD, and KTS acknowledge CAPES (Coordination for the Improvement of High Education Personnel) for the scholarship. Finally, we also thank the professor Luiz Carlos Pinheiro Machado Filho, PhD for all the shared knowledge that motivated us to work with social behavior.”

“We thank the São Paulo Research Foundation (FAPESP -  https://fapesp.br/en; grant #2017/10630-2 and grant #2017/50339-5).

JPSV, MD, and KTS acknowledge the Coordination for the Improvement of High Education Personnel (CAPES - https://www.iie.org/programs/capes ) for the scholarship.”

4. In your Data Journal Requirements: Availability statement, you have not specified where the minimal data set underlying the results described in your manuscript can be found. PLOS defines a study's minimal data set as the underlying data used to reach the conclusions drawn in the manuscript and any additional data required to replicate the reported study findings in their entirety. All PLOS journals require that the minimal data set be made fully available. For more information about our data policy, please see http://journals.plos.org/plosone/s/data-availability.

Additional Editor Comments:

Dear Authors,

The manuscript have certain suggestions from the reviewers that must be addressed before the final decision. Please respond to the comments of the reviewers and upload a revised version of your manuscript. Thank you.

Reviewers' comments:

Reviewer's Responses to Questions

**Comments to the Author**

1. Is the manuscript technically sound, and do the data support the conclusions?

Reviewer #1: Partly

Reviewer #2: Partly

2. Has the statistical analysis been performed appropriately and rigorously? 

Reviewer #1: N/A

Reviewer #2: N/A

3. Have the authors made all data underlying the findings in their manuscript fully available?

Reviewer #1: No

Reviewer #2: No

4. Is the manuscript presented in an intelligible fashion and written in standard English?

Reviewer #1: Yes

Reviewer #2: Yes

5. Review Comments to the Author

Reviewer #1: The paper presents an interesting approach as an R package to evaluate social hierarchy. However, there are aspects of that could be improved. The example lacks on detail, which makes the reader hard to follow. In addition, the intervals (line 121) seem critical to evaluate the social hierarchy, and the references used are from Holstein dairy cows and the authors mentioned that there is an effect of the species and category, so what happened with the example and the intervals used- where no Holstein cows? what happens when there are more than 2 animals – no dyadic? I suggest adding a section with the limitations of the package. The text will benefit to have a better explanation of the references cited (lines 157-167), is very hard to follow the idea if there is not a little more elaboration of the cited literature.

Specific comments:

L25: Farm animal welfare instead of the welfare of farm animals.

L30: creating instead of crated

L55: Replacement need a definition

L58: what does flexible means?

Table 1: what happens when there are more than 2 animals?

L 80: Bins

L101-102: sentence is not clear

L115: defined – describing

L125: This need to be clarified- there is an effect of species and category, but you run an example in Nellore cattle using intervals from Holstein?

L157-167: the methods mentioned here need a brief explanation as least

L210: to the dominance

L214: what does SH stands for?

Reviewer #2: The manuscript describes the existence and basic use of the 'socialh' R package.

In general, the manuscript is well written. However, there are many minor English language usage issues throughout the text. These could probably be remedied easily by having a native speaker proofread the text.

The software seems reasonable as a starting point. It is somewhat rudimentary in its implementation, however, and lacks the following basic pieces:

* Included sample data sets

* Package vignette

* Fleshed-out help functions

The package should be considered as alpha-level at this point.

In fact, the paper as it currently stands is really only at the level of an first-draft of a package vignette. It does not rise to the level needed to publish in PLOS ONE.

To make this paper publishable in PLOS ONE, I think that most of the following are needed:

* Demonstrate or add at least some capability to perform visualization of the results.

* Demonstrate or add at least some capability to perform statistical analysis.

For example, either implement statistical theory around the Landau index or the improved index. This could be done without much effort using the bootstrap, probably, but perhaps there is actually some theoretical work around these statistics.

* Perform at least three analyses of data that are novel in some way. This could be analyzing datasets that the authors have available, or data from other researchers interested.

* Or, alternatively, using data from the literature, perform analyses that add to the understanding of the data or that amplify, verify, or contradict the already published analyses.

* Demonstrate the utility for large databases that is claimed in Line 285. A large database these days is probably on the order of 10 million records or more.

* Optionally, add additional linearity evaluations and provide some empirical evaluation of them via simulation, followed by an evaluation of several real data sets.

* Optionally, perform a set of simulations that adds to the ability of researchers to meaningfully choose among ranking, or provides some sort of empirical insight into the use of the software, or provides some empirical validation of the interpretation of the outputs.

For future work, it might be useful to consider the following:

* Generalize the use of methodology to determine dominance.

That is, allow the user to specify different methodologies for this step. The authors list several, but hard-code only one.

Flexibility could be achieved by using a function name as an input parameter, for example, where the function is expected to follow a specific API. That would allow a user to program their own methodology quite easily. This would avoid the authors' having to provide a hard-coded list of options. R provides other methods for increasing flexibility, this is only one suggestion.

This would allow demonstrating empirically that different interpretations using different dominance evaluations, for example.

* Similarly, make social rank a user-definable set of cut-points, rather than hard-coded as only 3 values and 3 labels, and at the default cut-points provided by the R cut() function.

At a minimum, here, define these in the start of the function.

* In general, avoid any hard-coding of parameters inside the functions.

Make these user-definable at best or define them using variables at the start of the function at worst. At least the latter prepares the code for future generalization and makes coding decisions more visible.

* If visualization methods are constructed, add ggplot interfaces so that users can automatically have access to a flexible graphical presentation system.

6. PLOS authors have the option to publish the peer review history of their article (what does this mean?). If published, this will include your full peer review and any attached files.

Reviewer #1: No

Reviewer #2: No

---

## [Author Response · Author response to Decision Letter 0]

25 Apr 2022

We have attached the file of "responses of review" with the answer of all comments.

---

## [Decision Letter · Decision Letter 1]

9 May 2022

PONE-D-21-29427R1socialh : An R package for determining the social hierarchy of animals using data from individual electronic binsPLOS ONE

Dear Dr. Valente,

Thank you for submitting your manuscript to PLOS ONE. After careful consideration, we feel that it has merit but does not fully meet PLOS ONE’s publication criteria as it currently stands. Therefore, we invite you to submit a revised version of the manuscript that addresses the points raised during the review process. I invite you to revise your manuscript based on reviewer#2 feedback. Please note that a reviewer provided extensive editorial corrections and suggestions to the text, and I believe you should take that into account when revising your manuscript. Thanks.

We look forward to receiving your revised manuscript.

Kind regards,

Arda Yildirim, Ph.D.

Academic Editor

PLOS ONE

Journal Requirements:

Additional Editor Comments:

Dear authors;

Reviewer#2 have expressed positive feedback and important comments and suggestions on various aspects of your study. I concur that the study has merit, but before a final recommendation by the PLOS ONE can be made, I invite you to revise your manuscript based on reviewer#2 feedback. Please note that a reviewer provided extensive editorial corrections and suggestions to the text, and I believe you should take that into account when revising your manuscript. Thanks.

Reviewers' comments:

Reviewer's Responses to Questions

**Comments to the Author**

1. If the authors have adequately addressed your comments raised in a previous round of review and you feel that this manuscript is now acceptable for publication, you may indicate that here to bypass the “Comments to the Author” section, enter your conflict of interest statement in the “Confidential to Editor” section, and submit your "Accept" recommendation.

Reviewer #1: All comments have been addressed

Reviewer #2: (No Response)

2. Is the manuscript technically sound, and do the data support the conclusions?

Reviewer #1: Yes

Reviewer #2: Yes

3. Has the statistical analysis been performed appropriately and rigorously? 

Reviewer #1: N/A

Reviewer #2: N/A

4. Have the authors made all data underlying the findings in their manuscript fully available?

Reviewer #1: Yes

Reviewer #2: No

5. Is the manuscript presented in an intelligible fashion and written in standard English?

Reviewer #1: Yes

Reviewer #2: Yes

6. Review Comments to the Author

Reviewer #1: The authors have addressed all the comments and with that the paper has improve and therefore I considered that can be publish.

Reviewer #2: Although the authors have made some additions to their manuscript, the overall level of the manuscript is still somewhat low. While the implementation of the calculations in R is useful, it is not necessarily difficult or groundbreaking; nor do there appear to be any surprising results or new insights shown or described.

A quick glance at the following shows that they may be more appropriate places to publish for the current level of the manuscript:

* The Journal of Statistical Software <https: index="" www.jstatsoft.org="">

* The R Journal <https: journal.r-project.org="">

These journals are specifically geared toward providing a place to publish these types of implementations. The Journal of Statistical Software arguably tends toward more sophisticated analyses than the R Journal.

It still seems necessary to demonstrate some overall usefulness of this new package in at least two more situations.

An alternative would be some simulation studies using the package that could shed light on the capabilities or limitations of the methodology.

It would be quite simple to implement bootstrap statistical tests to produce confidence limits and/or tests of point hypotheses.

To summarize:

* The package implementation is useful, but on its own is more suited to another journal setting.

* One example of an analysis is interesting but does not seem to yield any insight or new results.

* Perhaps performing two or more analyses could allow the authors to draw some conclusion about the methodology.

* Perhaps using simulation studies the authors could derive some conclusion about the methodology.

* It would be easy to implement some simple statistical methods --- maybe these would produce some interesting results for either this data set or others.

* Perhaps there is deeper statistical theory already extant that could be incorporated.

A few minor comments:

* Line 345-346 Again, these are simply not large databases, especially for the simple calculations performed by the software.

* Understanding that the parameters are already defined for the functions, it is still best to make them user-definable in general. This can be easily accomplished using default values, so the user never needs to set them if that is preferred.</https:></https:>

7. PLOS authors have the option to publish the peer review history of their article (what does this mean?). If published, this will include your full peer review and any attached files.

Reviewer #1: No

Reviewer #2: No

---

## [Author Response · Author response to Decision Letter 1]

23 Jun 2022

We thank the editor and the reviewer for all contributions to our manuscript. We are grateful for the reviewer’ feedback, which helped improve the quality of the manuscript. We have answered to all comments and requests made by the reviewer. The changes in the manuscript are highlighted in yellow. The details of the responses are given bellow each comment in the "Response to Reviewers" file.

---

## [Decision Letter · Decision Letter 2]

29 Jun 2022

socialh : An R package for determining the social hierarchy of animals using data from individual electronic bins

PONE-D-21-29427R2

Dear Dr. Valente,

We’re pleased to inform you that your manuscript has been judged scientifically suitable for publication and will be formally accepted for publication once it meets all outstanding technical requirements.

Kind regards,

Arda Yildirim, Ph.D.

Academic Editor

PLOS ONE

Additional Editor Comments (optional):

Many thanks for sincerely and thoroughly considering and attending to the comments and concerns. Regards,

Reviewers' comments:

Reviewer's Responses to Questions

**Comments to the Author**

1. If the authors have adequately addressed your comments raised in a previous round of review and you feel that this manuscript is now acceptable for publication, you may indicate that here to bypass the “Comments to the Author” section, enter your conflict of interest statement in the “Confidential to Editor” section, and submit your "Accept" recommendation.

Reviewer #2: All comments have been addressed

2. Is the manuscript technically sound, and do the data support the conclusions?

Reviewer #2: (No Response)

3. Has the statistical analysis been performed appropriately and rigorously? 

Reviewer #2: (No Response)

4. Have the authors made all data underlying the findings in their manuscript fully available?

Reviewer #2: (No Response)

5. Is the manuscript presented in an intelligible fashion and written in standard English?

Reviewer #2: (No Response)

6. Review Comments to the Author

Reviewer #2: The authors have clarified some points, have provided several extensions to the analysis, and have provided more context in the form of real-world data analysis that should make it more suitable for the PLOS ONE audience. My apologies for missing that the data were in fact available.

7. PLOS authors have the option to publish the peer review history of their article (what does this mean?). If published, this will include your full peer review and any attached files.

Reviewer #2: No

---

## [Editor Report · Acceptance letter]

5 Jul 2022

PONE-D-21-29427R2 

socialh: An R package for determining the social hierarchy of animals using data from individual electronic bins 

Dear Dr. Valente:

I'm pleased to inform you that your manuscript has been deemed suitable for publication in PLOS ONE. Congratulations! Your manuscript is now with our production department. 

Kind regards, 

on behalf of

Prof. Dr. Arda Yildirim 

Academic Editor

PLOS ONE